# Antioxidant Effects and Phytochemical Properties of Seven Taiwanese *Cirsium* Species Extracts

**DOI:** 10.3390/molecules26133935

**Published:** 2021-06-28

**Authors:** Zi-Wei Zhao, Hung-Chi Chang, Hui Ching, Jin-Cherng Lien, Hui-Chi Huang, Chi-Rei Wu

**Affiliations:** 1The Institute of Basic Medical Sciences, National Cheng Kung University, Tainan 701, Taiwan; wei111783@gmail.com; 2Department of Golden-Ager Industry Management, College of Management, Chaoyang University of Technology, Taichung 413, Taiwan; changhungchi@cyut.edu.tw; 3Department of Pharmacy, Taichung Hospital, Ministry of Health and Welfare, Taichung 404, Taiwan; taic73047@taic.mohw.gov.tw; 4School of Pharmacy, China Medical University, Taichung 404, Taiwan; jclien@mail.cmu.edu.tw; 5Department of Chinese Pharmaceutical Sciences and Chinese Medicine Resources, College of Pharmacy, China Medical University, Taichung 404, Taiwan; hchuang@mail.cmu.edu.tw

**Keywords:** *Cirsium* species, silymarin, silicristin, antioxidant activities, phytochemical profiles, HPLC–DAD

## Abstract

In the present investigation, we compared the radical-scavenging activities and phenolic contents of seven Taiwanese *Cirsium* species with a spectrophotometric method. We further analyzed their phytochemical profiles with high-performance liquid chromatography–photodiode array detection (HPLC–DAD). We found that the flower part of *Cirsium japonicum* var. *australe* (CJF) showed the best radical-scavenging activities against 1,1-diphenyl-2-picrylhydrazyl (DPPH), 2,2′-azino-bis(3-ethylbenzothiazoline-6-sulfonic acid) (ABTS), and the hypochlorite ion, for which the equivalents were 6.44 ± 0.17 mg catechin/g, 54.85 ± 0.66 mmol Trolox/g and 418.69 ± 10.52 mmol Trolox/g respectively. CJF also had the highest contents of total phenolics (5.23 ± 0.20 mg catechin/g) and phenylpropanoids (29.73 ± 0.72 mg verbascoside/g). According to the Pearson’s correlation coefficient, there was a positive correlation between the total phenylpropanoid content and ABTS radical-scavenging activities (r = 0.979). The radical-scavenging activities of the phenylpropanoids are closely related to their reducing power (r = 0.986). HPLC chromatograms obtained in validated HPLC conditions confirm that they have different phytochemical profiles by which they can be distinguished. Only CJF contained silicristin (0.66 ± 0.03 mg/g) and silydianin (9.13 ± 0.30 mg/g). CJF contained the highest contents of apigenin (5.56 ± 0.09 mg/g) and diosmetin (2.82 ± 0.10 mg/g). Among the major constituents, silicristin had the best radical-scavenging activities against DPPH (71.68 ± 0.66 mg catechin/g) and ABTS (3.01 ± 0.01 mmol Trolox/g). However, diosmetin had the best reducing power and radical-scavenging activity against the hypochlorite anion (41.57 ± 1.14 mg mmol Trolox/g). Finally, we found that flavonolignans (especial silicristin and silydianin) and diosmetin acted synergistically in scavenging radicals.

## 1. Introduction

The genus *Cirsium* (common thistle) comprises perennial, biennial, or, rarely, annual spiny species from the family *Asteraceae*. According to the collection of The Plant List and Plants of the World online in the Royal Botanic Gardens of Kew, there are about 450–480 accepted species in the genus [1,2]. They are native plants distributed in the northern hemisphere including Eurasia, North Africa and North America [3]. In Japan, there are about 120 native *Cirsium* species [4]. According to the Flora of China, there are about 50 native *Cirsium* species in China [5]. In the Flora of Taiwan, 10 *Cirsium* species are described, including *C. arisanense* Kitam. (CA), *C. brevicaule* A. Gray (CB), *C. ferum* Kitam. (CF), *C. hosokawae* Kitam. (CHO), *C. japonicum* DC. var. *australe* Kitam. (CJ), *C. japonicum* DC. var. *takaoense* Kitam. (CJT), *C. kawakamii* Hayata (CK), *C. lineare* (Thunb.) Sch. Bip. (CL), *C. morii* Hayata (CM), and *C. suzukii* Kitam. (CS) [6]. Some researchers have compared the characteristics or phylogenetic efficacy of different *Cirsium* species from local areas or some *Cirsium* species from different areas [7,8,9,10,11]. Additionally, researchers have constructed the phytochemical fingerprints of some *Cirsium* species with thin-layer chromatography (TLC) or high-performance liquid chromatography (HPLC) [12,13]. However, there is very little literature comparing Taiwanese *Cirsium* species apart from our previous report [14].

In Taiwan, the genus *Cirsium* is called “Formosan thistle” and is commonly used as a folk medicine to treat alcoholic liver disease. Ku et al. [15] indicated that the phenol-containing aqueous components from the leaves/stems of CA protected against tacrine-induced hepatotoxicity. Our previous study found that the flower part of CJ most strongly protected against acute CCl_4_-induced hepatic damage among the three most common *Cirsium* species (CA, CK, and CJ) and a commercial *Cirsium* product—Cirsii Herba (CH) [14]. These hepatoprotective effects are mostly mediated by their antioxidant activities [14,15]. Hence, the first purpose of this study was to compare the radical-scavenging activities of the aerial (A), radix (R), and flower (F) parts of seven *Cirsium* species (CA, CB, CF, CHO, CJ, CK, and CL) from the fields in Taiwan against the 1,1-diphenyl-2-picrylhydrazyl (DPPH) radical, 2,2′-azino-bis(3-ethylbenzothiazoline-6-sulfonic acid) (ABTS) radical, and hypochlorite ion (ClO^−^) with a microtiter spectrophotometric method in vitro. Their radical-scavenging activities were also compared with those of a commercial *Cirsium* product—CH. The genus *Cirsium* is one of the sources of “thistles” and possesses similar pharmacological activities to milk thistle. Silymarin, a standardized preparation extracted from the seeds of milk thistle, is one of the most prescribed natural botanical preparations. Silymarin also exerts hepatoprotective effects via its antioxidative activity, its hepatoprotective effect is mainly used for various types of liver disease, such as alcoholic liver disease, non-alcoholic fatty liver disease, drug- and toxin-induced liver disease, cholestasis, and liver fibrosis [16,17,18,19]. The major ingredients of silymarin are the phenolic compounds, including 70–80% (*w*/*w*) flavonolignans and flavonoids. The silymarin flavonolignans are mainly composed of 50–70% silibinin (α and β), isosilibinin (α and β), silydianin, and silicristin [17,20,21,22]. Silymarin and these above flavonolignans function as natural antioxidants and antidotes, protecting against different biological (mycotoxins, snake venoms, and bacterial toxins) and chemical (metals, fluoride, pesticides, cardiotoxins, neurotoxins, hepatotoxins, and nephrotoxins) poisons [17,21,23]. Our earlier report indicated that the hepatoprotective ingredients of CJF might be flavonolignans (silibinin and silydianin) and flavonoids (diosmetin) [14]. Hence, the second purpose of this study was to further evaluate the radical-scavenging activities of all the ingredients in the above-mentioned *Cirsium* species and their synergic effects. Silymarin was used as a positive control.

In a series study on the antioxidant activities of the inflorescences and leaves of *Cirsium* species produced in the northeast region of Poland, it was found that their antioxidant activities were positively correlated with the total phenolic content [24,25]. According to Kim et al. [11], the products formed through the phenylpropanoid metabolic pathway are the major phytochemical ingredients of *Cirsium*. Our earlier study indicated that the hepatoprotective ingredients of CJF might be flavonolignans (silibinin and silydianin) and flavonoids (diosmetin) [14]. However, the antioxidant ingredients and HPLC fingerprints of other Taiwanese *Cirsium* species have not been investigated and constructed. Hence, a third objective was to measure the content of the phenolic compounds (including the total phenolics, phenylpropanoids, and flavonoids) in the above-mentioned *Cirsium* species using a microtiter spectrophotometric assay. According to our previous report and other phytochemical reports, the phytochemical profiles (flavonolignans and flavonoids; their structures are shown in Figure 1) and antioxidant fingerprints of these *Cirsium* species were further analyzed with HPLC–photodiode array detection (HPLC–DAD).

## 2. Results

### 2.1. The Contents of Total Phenolics, Flavonoids and Phenylpropanoids

The contents of the total phenolics, flavonoids, and phenylpropanoids in the methanolic standardized extracts of *Cirsium* species and CH are presented in Table 1, which were measured using 96-well microtiter spectrophotometric methods. CJF has the highest contents of total phenolics and total phenylpropanoids among all the *Cirsium* species and CH. The next top five in order of the contents of total phenolics in other *Cirsium* species and CH are CLA > CFA > CLF > CAA > CKA. The lowest content of total phenolics was found in CJR. The next top five in order of the contents of total phenylpropanoids in other *Cirsium* species and CH are CFA > CLA > CLF > CAA > CKA. The lowest content of total phenylpropanoids found in CJA. CLA has the highest content of total flavonoids among all *Cirsium* species and CH. The next top five in order of the contents of total flavonoids in other *Cirsium* species and CH are CFA > CLF > CAA > CFF > CBA. The lowest content of total flavonoids is found in CAF.

### 2.2. Validation of the HPLC–DAD Method and the Phytochemical Profiles of Cirsium Species

Furthermore, we selected the nine methanolic standardized extracts of *Cirsium* species with higher antioxidant phytochemical content and CH to analyze the phytochemical profiles by HPLC–DAD. The HPLC chromatographs of the standards and the methanolic standardized extracts of CJF and CH are shown in Figure 2. The HPLC chromatographs of the standards and the methanolic standardized extracts of other *Cirsium* species are shown in Appendix A. We confirmed the specificity of the peaks for the methanolic standardized extracts of *Cirsium* species by comparing their retention times and UV spectra with those of all the standards. The well-resolved peaks from the HPLC chromatograms indicated the reasonable analytical specificity of this HPLC method. Hence, this HPLC–DAD method could be used to identify the Taiwanese *Cirsium* species and commercial *Cirsium* material, CH, because there are different phytochemical profiles for the nine methanolic standardized extracts of the *Cirsium* species and CH.

The calibration curves of all the standards showed excellent linear regression with linear correlation coefficients (r^2^) greater than 0.998 over the working range 5.0–20.0 μg/mL, as shown in Table 2. These results were better than our previous findings [14], in which the correlation coefficients of the calibration plots for apigenin, diosmetin, silibinin α, silibinin β, silydianin, silicristin, isosilibinin α, and isosilibinin β were 0.992–0.994. The reliably detected (LOD) and quantified (LOQ) minimum concentration levels of all the standards were 0.31–1.33 μg/mL and 0.93–4.00 μg/mL, respectively (Table 2). As a criterion for precision, the degree of agreement (scatter) among the measured values for the same samples is important for the HPLC method. To verify the repeatability, three different concentrations of the mixed standard were analyzed. The residual standard deviations (RSDs, %) were calculated from the results of repeated measurements. The analysis was found to be precise, with intraday and interday variability (% RSD) for the peak areas of all the standards in the ranges of 0.27–2.23% and 1.66–2.98%, respectively (Table 3).

The contents for the phytochemical profiles of the nine methanolic standardized extracts of *Cirsium* species and CH are presented in Table 4. Only CJF contained silicristin (0.66 ± 0.03 mg/g) and silydianin (9.13 ± 0.30 mg/g). In addition, CJF also contained silibinins α (0.48 ± 0.04 mg/g) and β (1.11 ± 0.14 mg/g), apigenin (5.56 ± 0.09 mg/g), and diosmetin (2.82 ± 0.10 mg/g). The contents of apigenin and diosmetin in CJF are higher than in the other Taiwanese *Cirsium* species and the commercial *Cirsium* material, CH. Only CLF contained isosilibinin β (0.33 ± 0.01 mg/g). CLF also contained apigenin (0.62 ± 0.02 mg/g) and diosmetin (1.71 ± 0.05 mg/g). CH and every part of CF only contained silibinins α and β. In particular, CFA had the highest contents of silibinins α and β among all the Taiwanese *Cirsium* species and the commercial *Cirsium* material, CH. However, CAA only contained silibinin α (0.76 ± 0.01 mg/g). CBA and CKR only contained silibinin β (0.34 ± 0.05 and 0.24 ± 0.00 mg/g, respectively). CLA only contained diosmetin (1.39 ± 0.01 mg/g).

### 2.3. DPPH-Radical Scavenging Capacity

The IC_50_ for catechin’s scavenging of the DPPH radical was 25.46 ± 0.28 μg/mL. The relative DPPH-radical-scavenging activities (CEDSC values) of the methanolic standardized extracts of *Cirsium* species and CH and their ingredients were obtained by calculating the IC_50_ values for the above *Cirsium* samples and the IC_50_ values for the positive control catechin. The CEDSC values of the methanolic standardized extracts of *Cirsium* species and CH and their ingredients are presented in Figure 3. Among the methanolic standardized extracts of *Cirsium* species and CH, CJF showed the highest DPPH-radical-scavenging capacity. The next top five in order of DPPH-scavenging activities in the other *Cirsium* species and Cirsii Herb is CLA > CFA > CAA > CLF > CKR. The lowest DPPH scavenging activity was observed for CHOA (Figure 3a).

Among all the ingredients of the methanolic standardized extracts of the *Cirsium* species and CH, silicristin showed the highest DPPH-radical-scavenging capacity but its potency was lower than that of silymarin. The next ingredients in order of DPPH-scavenging activities were silydianin > diosmetin > isosilibinin > silibinin. The DPPH-scavenging percentage of apigenin at the highest concentration (1 mg/mL) used in this experiment did not reach 50% (Figure 3b). According to the contents of the major phytochemical ingredients obtained from HPLC–DAD, the CJF methanolic standardized extract, at the IC_50_ for scavenging the DPPH radical contained 2.59 μg of silicristin, 33.65 μg of silydianin, and 10.35 μg of diosmetin. At these concentrations, the scavenging percentages of these above ingredients for the DPPH radical were 7.52 ± 1.59, 13.06 ± 1.52, and 3.89 ± 0.40, respectively. The combination of the above ingredients according to the contained proportions could eliminate 47.14 ± 1.67% of DPPH radicals.

### 2.4. ABTS-Radical Scavenging Capacity

The IC_50_ value of Trolox for scavenging the ABTS radical is 131.58 ± 2.41 μM. The relative ABTS radical-scavenging activities (TEAC values) of the methanolic standardized extracts of *Cirsium* species and CH and their ingredients were obtained by calculating the IC_50_ values for the above *Cirsium* samples and the IC_50_ values for the positive control, Trolox. The TEAC values of the methanolic standardized extracts of *Cirsium* species and CH and their ingredients are presented in Figure 4. Among the methanolic standardized extracts of *Cirsium* species and CH, CJF also showed the highest ABTS radical-scavenging capacity. The next top five in order of ABTS scavenging activities in the other *Cirsium* species and CH were CLA > CFA > CAA > CLF > CAR. The lowest ABTS-scavenging activity was observed for CHOA (Figure 4a).

Among all the ingredients of the methanolic standardized extracts of *Cirsium* species and CH, silicristin showed the highest ABTS-radical-scavenging capacity and its potency was greater than that of silymarin. The next ingredients in order of ABTS-scavenging activities were isosilibinin > silibinin > silydianin > diosmetin > apigenin (Figure 4b). According to the contents of major phytochemical ingredients obtained from HPLC–DAD, the CJF methanolic standardized extract, at the IC_50_ for scavenging the ABTS radical, contained 2.40 μg of silicristin, 31.23 μg of silydianin, and 9.61 μg of diosmetin. At these concentrations, the scavenging percentages for these above ingredients for the ABTS radical were 4.18 ± 0.25, 9.98 ± 0.21, and 3.57 ± 0.56, respectively. The combination of the above ingredients according to the contained proportions could eliminate 40.25 ± 0.20% of ABTS radicals.

### 2.5. Hypochlorite-Ion-Scavenging Assay

The IC_50_ value of Trolox for scavenging the hypochlorite ion is 221.22 ± 3.14 μM. The relative hypochlorite-ion-scavenging activities (Trolox equivalent values) of the methanolic standardized extracts of *Cirsium* species and CH and their ingredients were obtained by calculating the IC_50_ values for the above *Cirsium* samples and the IC_50_ values for the positive control, Trolox. The hypochlorite-ion-scavenging activities of the methanolic standardized extracts of *Cirsium* species and CH and their ingredients are presented in Figure 5. Among the methanolic standardized extracts of *Cirsium* species and CH, CJF also showed the highest hypochlorite-ion-scavenging capacity. The next top five in order of hypochlorite-ion-scavenging activities in the other *Cirsium* species and Cirsii Herb were CFA > CLA > CAA > CKA > CFF. CJA showed the lowest hypochlorite-ion-scavenging activity (Figure 5a).

Among all the ingredients of the methanolic standardized extracts of *Cirsium* species and CH, diosmetin showed the highest hypochlorite-ion-scavenging activity, and it was more potent than silymarin. The next ingredients in order of hypochlorite-ion-scavenging activities were isosilibinin > silydianin > silicristin > silibinin. The hypochlorite-ion-scavenging percentage for apigenin at the highest concentration (1 mg/mL) used in this experiment did not reach 50% (Figure 5b). According to the contents of major phytochemical ingredients obtained from HPLC–DAD, CJF at the IC_50_ for scavenging the hypochlorite ion contained 0.32 μg of silicristin, 4.19 μg of silydianin, and 1.29 μg of diosmetin. At these concentrations, the scavenging percentages for these ingredients against the hypochlorite ion are 4.51 ± 0.82, 19.05 ± 0.31, and 13.38 ± 0.20, respectively. The combination of the above ingredients according to the contained proportions could eliminate 47.60 ± 0.60% of hypochlorite ions.

### 2.6. Ferric-Reducing-Antioxidant Power (FRAP) Assay

The reaction slope of Trolox at 17.5–300 μM for the ferric reduction reaction is 2.95 ± 0.04. The relative FRAP values (Trolox equivalent values) of the methanolic standardized extracts of *Cirsium* species and CH and their ingredients were obtained by calculating the reaction slope for the above *Cirsium* samples and the reaction slope for the positive control, Trolox. The FRAP values of the methanolic standardized extracts of *Cirsium* species and CH and their ingredients are presented in Figure 6. Among the methanolic standardized extracts of *Cirsium* species and CH, CJF also showed the highest ferric-reducing/antioxidant power. The next top five in order of ferric-reducing/antioxidant power in other *Cirsium* species and Cirsii Herb is CFA > CLA > CLF > CAA > CKR. CJA showed the lowest ferric-reducing/antioxidant power (Figure 6a).

Among all the ingredients of the methanolic standardized extracts of *Cirsium* species and CH, diosmetin showed the highest ferric-reducing/antioxidant power, with a potency greater than that of silymarin. The next ingredients in order of ferric reducing/antioxidant power were silicristin > silibinin > isosilibinin > silydianin > apigenin (Figure 6b).

## 3. Discussion

There are about 480 species of the genus *Cirsium* in the northern hemisphere of the world. Among them, some native *Cirsium* species are used to treat hemorrhage, edema, nephritis, and hepatitis by local traditional physicians. Recent studies have pointed out that these *Cirsium* species possess anticancer, antidiabetic, hepatoprotective, renoprotective, and neuroprotective effects through their antioxidant activities [26,27,28,29,30,31,32,33,34,35]. Our previous report also found that three Taiwanese *Cirsium* species could protect against acute CCl_4_-induced hepatic damage through their antioxidant activities [14]. In 2008, Nazaruk compared the total phenolic content and antioxidant activities of the inflorescences and leaves of five native Poland *Cirsium* species using Folin–Ciocalteau reagent and the ABTS-radical method [24]. In addition to the ABTS-radical method, the DPPH-radical method is one of the simplest and quickest methods used, with dozens of in vitro models for assaying the antioxidant activities of plant extracts and phenolic compounds. We found that the A, F, and R parts of seven *Cirsium* species in Taiwan had varying degrees of antioxidant activities against both free radicals. Among the sixteen methanolic standardized extracts of Taiwanese *Cirsium* species and a traditional Chinese product (CH), CJF exhibited the best antioxidant activities according to the ABTS and DPPH methods. CLA, CFA, and CAA had the second-, third-, and fourth-best antioxidant activities, respectively. CHOA showed the worst antioxidant activity.

Hypochlorous acid (HOCl), a powerful antimicrobial agent, is generated from the oxidation of hydrogen peroxide (produced during respiratory burst) and chloride catalyzed by a neutrophil-derived heme peroxidase-myeloperoxidase (MPO). Due to its high reactivity and easy dissociation, HOCl and its conjugate base (OCl^−^) are potent oxidants that react with a number of biomolecules of the host’s tissues and cells, such as proteins, lipids, nucleotides, and DNA. Proteins (as well as peptides and amino acids) are major targets for HOCl. HOCl can convert tyrosine residues and histidine and lysine side chains in a protein to form chloramines via chlorine-transfer reactions. HOCl oxidizes NADH and the NH groups of pyrimidine nucleotides, cytidine monophosphate, and adenosine monophosphate via chlorination. HOCl also chlorinates the vinyl ethers of lipids to form α-chlorofatty aldehyde and the unsaturated molecular species of lysophosphatidylcholine. In addition, it attacks the double bonds of unsaturated fatty acids and cholesterol and also causes the extensive denaturation of double-stranded DNA. Furthermore, HOCl can react with other compounds to produce other reactive oxidant species (ROS). For example, HOCl can react with the superoxide anion to generate the hydroxyl radical. Its conjugated base, the hypochlorite ion, can react with hydrogen peroxide to produce singlet oxygen species. These ROS subsequently generated from HOCl can cause further cellular damage. Hence, HOCl and its conjugated base, the hypochlorite ion, participate in many pathophysiological diseases, including inflammatory diseases, cardiovascular diseases, hepatitis, glomerulonephritis, obesity, diabetes, and cancer. The present study further found that the A, F, and R parts of seven *Cirsium* species in Taiwan have varying degrees of antioxidant activities against the hypochlorite ion. The four with the greatest antioxidant activities against the hypochlorite ion were CJF, CFA, CLA, and CAA. In terms of the parts of *Cirsium* species, the A part is usually the most active in antioxidant activity, except for in CJ. Then, CF, CL, and CA are the three best species in terms of the A parts of seven Taiwanese *Cirsium* species. The above three *Cirsium* species were collected from high-altitude areas in Taiwan.

Our previous report and Nazaruk’s report [24,36] indicated that the phenolic compounds (such as phenolic acids, phenylpropanoids, and flavonoids) possessed antioxidant activities and that there was a close linear correlation between the total phenolic contents and antioxidant activities. Flavonoids are the most widespread and diverse phenolic compounds in plants [37]. Phenylpropanoids are intermediate compounds in the biosynthesis of flavonoids and some phenolic compounds (phenylpropanoid metabolic pathway) [38]. Moreover, the products (mainly including flavonoids) biosynthesized through the phenylpropanoid metabolic pathway are the major phytochemical ingredients of the genus *Cirsium* according to Kim et al. [11]. We also found, using the Folin–Ciocalteau and Arnow methods, that CJF had the highest contents of total phenolics and total phenylpropanoids among the sixteen methanolic standardized extracts of Taiwanese *Cirsium* species and a traditional Chinese material, CH. Expect for CJF, CFA, CLA, and CAA had the highest contents of total phenolics and total phenylpropanoids. The contents of total phenolics in CJR were the lowest; however, CJA had the lowest contents of total phenylpropanoids. CLA had the highest contents of total flavonoids among the sixteen methanolic standardized extracts of Taiwanese *Cirsium* species and CH. In terms of the parts of the genus *Cirsium* except for CJ, the A part usually showed the highest contents of total phenolics, total flavonoids, and total phenylpropanoids. CF, CL, and CA, in the A parts of seven Taiwanese *Cirsium* species, showed higher content of phenolic ingredients. Many reports indicated that the antioxidant activities of plant extracts, vegetables, and fruits are closely related to their reducing power [36]. FRAP is a simple and rapid method for actually measuring the reducing capability of antioxidants and screening their ability to maintain the redox status in cells [39]. The present results show that CJF, CFA, CLA, and CAA have the greatest reducing capacities. In terms of the parts of genus *Cirsium* species except for CJ, the A part is also the most active in reducing power. CF, CL, and CA were the three best species regarding the A parts of seven Taiwanese *Cirsium* species. Hence, we carried out Pearson correlation coefficient analysis for the above measures of antioxidant potency (DPPH, TEAC, hypochlorite ion and FRAP) and the amounts of antioxidant ingredients in sixteen methanolic standardized extracts of Taiwanese *Cirsium* species and CH. Firstly, all the antioxidant assays (DPPH, TEAC, hypochlorite-ion radical, and FRAP) showed high correlation coefficients, suggesting the reliability and interchangeability of these spectrophotometer-based methods in predicting the antioxidant activities of plant extracts (Table 5). The DPPH-radial method is based on the reduction of the purple DPPH^•^ to 1,1-diphenyl-2-picryl hydrazine, mainly through both single-electron transfer (SET) and hydrogen-atom transfer (HAT) reactions. However, it is usually more biased towards the SET-based rather than HAT-based mechanism. The ABTS-radical method is based on the generation of a blue/green ABTS^•+^, which is decolorized by antioxidant compounds. This decolorization reaction mainly occurs through a HAT-based, rather than SET-based, mechanism. In general, DPPH must be dissolved in an organic solvent (such as methanol or ethanol); hence, it is applicable for hydrophobic antioxidant systems and sensitive to antioxidant compounds of low to medium polarity. ABTS is usually dissolved in an aqueous solution or ethanol, so it is applicable to both hydrophilic and hydrophobic antioxidants. The hypochlorite ion must be dissolved in an aqueous solution, and hence, it is applicable to hydrophilic antioxidants. This scavenging reaction occurs through a HAT-based, SET-based, or a mixed mechanism. The FRAP assay, different from the above three methods, involves the reduction of ferric iron (Fe^3+^) to ferrous iron (Fe^2+^). FRAP reagent must be dissolved in an aqueous solution, so it is applicable to hydrophilic antioxidants. The mechanism of the FRAP assay is mainly based on SET rather than HAT. Moreover, the redox potential of ferric iron (Fe^3+^) (−0.70 V) is close to that of ABTS^•+^ (0.68 V) [40,41]. Hence, our present results confirm the above argument that the values of the FRAP assay are much closer to the values of the ABTS method than those of the DPPH method, which is based on the characteristics of the solvent and the redox potential. Based on this phenomenon, we suggested that the antioxidant mechanism of the genus *Cirsium* might be based on a SET reaction rather than a HAT reaction. Furthermore, these results indicate a significant positive correlation between the above antioxidant potency and antioxidant ingredients, except for the total flavonoid content (Table 4). Mainly, the total phenylpropanoid content was positively and highly correlated with the DPPH-radical (r = 0.938), ABTS-radical (r = 0.979), and hypochlorite-ion (r = 0.906) scavenging capacity. The total phenolic content was also positively and highly correlated with the DPPH-radical (r = 0.940), ABTS-radical (r = 0.897), and hypochlorite-ion (r = 0.900) scavenging capacities. Additionally, FRAP was highly correlated with the total phenylpropanoid content (r = 0.980) and total phenolic content (r = 0.923). Based on the above statistical analyses, the findings of this investigation match those of other reports [24,36], that the antioxidant activities of plant extracts, vegetables, and fruits are closely related to their contents of phenolic ingredients. Our results further suggest that the antioxidant activities of sixteen methanolic standardized extracts of Taiwanese *Cirsium* species are more closely related to the content of total phenylpropanoids. These compounds in the methanolic standardized extracts of Taiwanese *Cirsium* species act as single-electron or -proton donors to provide good reducing power and antioxidant potency.

Nazaruk reported chlorogenic acid as a common antioxidant ingredient of five *Cirsium* species in Poland [24,42]. Ma et al. pointed out that flavonoids (luteolin) and flavonolignans (silibinin β) are the major antioxidant and hepatoprotective ingredients of Cirsii Japonici Herb (Japanese thistle) [26]. The major ingredients of *C. japonicum* var. *maackii* (Maxim.) Matsum. (Korean thistle) in terms of antioxidant and hepatoprotective effects are flavonoids (luteolin, apigenin, and their glucosides) [33]. The major ingredients of *C. setidens* Nakai (Korean thistle), in terms of antioxidant and hepatoprotective effects, are also flavonoids (pectolinarin and pectolinarigenin) [43]. Our previous report indicated that flavonoids (diosmetin) and flavonolignans (silibinin and silydianin) are the major hepatoprotective ingredients of CJF. According to the present HPLC–DAD results, we found that the phytochemical characteristics differed among nine methanolic standardized extracts of *Cirsium* species and CH. As in our previous report [14], CJF also contained silibinin, silydianin, and diosmetin. In addition, CJF contained silicristin and apigenin. However, only CJF contained silicristin and silydianin among all the Taiwanese *Cirsium* species. The contents of apigenin and diosmetin in CJF were the highest among the Taiwanese *Cirsium* species. CFA contained the most silibinins α and β among all the Taiwanese *Cirsium* species. This is consistent with the above results for total phenolic content. Hence, we suggested that flavonolignans are major ingredients of nine methanolic standardized extracts of Taiwan *Cirsium* species, although they possessed different phytochemical profiles. Flavonoids are also major ingredients of CJF and CLF.

Earlier reports indicated that flavonoids (apigenin and diosmetin) and flavonolignans (silibinin, silydianin, and silicristin) possessed antioxidant activities [44]. Our present results indicate that silicristin had the best antioxidant capacity against DPPH and ABTS radicals. Diosmetin had the best antioxidant capacity against the hypochlorite ion. Apigenin showed the weakest antioxidant capacity against any radical. These antioxidant results for flavonoids are the same as in other reports [45,46], in that the antioxidant potency of diosmetin according to the ABTS, DPPH, and FRAP methods is better than that of apigenin. However, these antioxidant results for flavonolignans according to the DPPH and TEAC assays differ in other reports [44]. This difference might be due to different evaluation methods (endpoint or kinetics in the DPPH assay) and reaction conditions (the reaction of ABTS with potassium persulfate or myoglobin in the TEAC assay). Moreover, we further found that the three major ingredients according to the contained proportions at the IC_50_ of CJF against three radicals only scavenge a small percentage (<20%) of three free radicals. The combination of the three major ingredients could eliminate about 50% (40–47%) of the three free radicals. Hence, the major antioxidant ingredients of CJF are flavonolignans (silicristin and silydianin) and flavonoids (diosmetin). In CJF, these act synergistically in scavenging free radicals. According to the present results and our previous report [14], we suggest that *Cirsium* species, especially CJF, contain similar ingredients and antioxidant activities as *Silybum marianum* and its standardized preparation, silymarin. Earlier reports indicated that the bioavailability of silibinins α and β, which account for about 50–70% of the silymarin mixture, is about 1% [44,47]. The orally effective dose of silibinin for protecting against hepatic damage is 100–200 mg/kg in rats, and for silymarin, it is 200 mg/kg [19,22]. According to the work of Valentova et al. [48], silymarin flavonolignans are resistant to the metabolic action of intestinal microbiota. Silymarin flavonolignans undergo extensive enterohepatic circulation, and phase I (demethylation) and phase II (glucuronidation and sulfation) of biotransformation by cytochrome P450 and UDP-glucuronosyltransferases (UGTs) [49]. Otherwise, another report indicated flavonolignans were biotransformed into demethyl derivatives by human intestinal bacteria *E. limosum* ZL-II, and these transformation products exhibited higher anti-Alzheimer’s activities than these flavonolignans [50]. However, a recent report indicated that the anti-Alzheimer’s activities of flavonolignans are partially related to regulating the gut microbiota community and bioactivity [51]. Although the contents of silibinins α and β in CJF are not high, our previous work indicated that the hepatoprotective effects of CJF at 0.5 mg/kg were better than those of silymarin (100 mg/kg) and equivalent to those of silibinin (25 mg/kg) [14]. Moreover, apigenin and diosmetin were absorbed quickly, had longer T_max_, and exhibited many biological activities [52]. Unabsorbed apigenin was metabolized into hydroxylate or ring-fission by-products by gut microbiota in the colon; however, it and its by-products may modify the function of gut microbiota, at least for *Enterococcus* spp. [53]. Diosmetin was less degraded by gut microbiota due to the presence of a methoxy group. Diosmetin mainly inhibited the growth of *Bacillus* spp. and *Helicobacter pylori* [54]. Based on these results and the related reports about the bioavailability and biological activity of related ingredients, it can be expected that Cirsium species, especially CJF, have similar pharmacological activities, such as hepatoprotection and the potential to be developed into functional products with biological activity for therapeutic use. As for the influence on the function and community of gut microbiota and the metabolism of *Cirsium* species and their ingredients, further investigation is needed.

## 4. Materials and Methods

### 4.1. Collection and Preparation of Plant Materials

Seven *Cirsium* species materials, including the aerial, flower, and radix parts of CA (CAA, CAF, and CAR); the aerial and radix parts of CB (CBA and CBR); the aerial and flower parts of CF (CFA and CFF); the aerial part of CHO (CHOA); the aerial, flower, and radix parts of CJ (CJA, CJF, and CJR); the aerial and radix parts of CK (CKA and CKR); and the aerial, flower, and radix parts of CL (CLA, CLF and CLR), were collected from four places in Taiwan (Alishan in Chiayi, Tianwei in Changhua, Yuanfeng in Nantou, and Taichung) from August to September 2017. The abbreviations and collection places for these *Cirsium* species and CH are presented in Table 6. These above materials were identified by Hung-Chi Chang of Chaoyang University of Technology in Taiwan. CH was obtained from the Herb Garden of China Medical University, Taichung City, Taiwan. All the parts of seven *Cirsium* species and CH (2.5 g) were ground and extracted with 20 mL of methanol by sonication for 90 min. Then, the resulting extracts were filtered with a 0.22 μm filter and were adjusted to volumes of 20 mL in order to obtain the methanolic standardized extracts of these *Cirsium* species and CH.

### 4.2. Chemicals

DPPH, ABTS, 2,4,6-tri(2-pyridyl)-s-triazine (TPTZ), 3,3′,5,5′-tetramethyl-benzidine (TMB), 6-hydroxy-2,5,7,8-tetramethy-chroman-2-carboxylic acid (Trolox), acetic acid, aluminum nitrate, (+)-catechin, ferric chloride, Folin-Ciocalteau reagent, potassium persulfate, quercetin, silibinin, silymarin, sodium carbonate, sodium hydroxide, sodium hypochlorite, sodium molybdate, and sodium nitrate were purchased from Sigma-Aldrich (St. Louis, MO, USA). Apigenin, diosmetin, and verbascoside were purchased from Extrasynthese (Genay, France). Isosilibin, silicristin, and silydianin were purchased from SunHank Technology Co., Ltd. (Tainan, Taiwan). Hydrogen chloride, HPLC-grade methanol, and *o*-phosphoric acid were purchased from Merck (Darmstadt, Germany).

### 4.3. Measuring Antioxidant Phytochemical Contents with a Spectrophotometric Reader

According to the 96-well microtiter spectrophotometric methods described in our previous report [36], we measured the contents of major antioxidant phytochemicals (including total phenolics, total flavonoids, and total phenylpropanoids) in the methanolic standardized extract of *Cirsium* species and CH with a microtiter spectrophotometric reader (PowerWave X340, Bio-Tek Inc., Winooski, VT, USA). Based on the redox reaction of phenolic compounds and Folin–Ciocalteau’s reagent, the contents of total phenolics in the methanolic standardized extracts of *Cirsium* species and CH are proportional to the absorbance (at 725 nm) of the formed, blue-colored product and can be represented as the amount relative to the positive control (+) catechin, which is expressed as milligrams of catechin equivalents per gram of the extract (mg catechin/g sample). Based on the aluminum complexation reaction of flavonoid compounds and aluminum chloride, the contents of total flavonoids in the methanolic standardized extracts of *Cirsium* species and CH are proportional to the absorbance (at 510 nm) of the formed colored product and can be represented by the amount relative to the positive control quercetin, which is expressed as milligrams of quercetin equivalents per gram of the extract (mg quercetin/g sample). The contents of total phenylpropanoids in the methanolic standardized extracts of *Cirsium* species and CH are proportional to the absorbance (at 525 nm) of the colored products formed from phenylpropanoids with Arnow reagent (containing 5% (*w*/*v*) sodium nitrate and 5% sodium molybdate) and are expressed as milligrams of verbascoside equivalents per gram of the extract (mg verbascoside/g sample).

### 4.4. Phytochemical Profiles Determined with HPLC–DAD

The phytochemical profiles of the methanolic standardized extracts of *Cirsium* species and CH were determined by HPLC–DAD, according to the chromatographic conditions described in our previous report [14]. Briefly, the Shimadzu HPLC VP series, a LiChrospher RP-18e (250 × 4 mm, 5 μm) column (Merck KGaA, Darmstadt, Germany), and Shimadzu Class-VP chromatography data systems were used. The separating conditions were also as described in our previous report [14]. The mobile phase was 0.5:35:65 phosphoric acid:methanol:water (solvent A) and 0.5:70:30 phosphoric acid:methanol:water (solvent B). The flow rate was modified to 0.8 mL/min. The gradient program conditions were modified as shown in Table 7. The stock solution (10 mg/mL) of each standard including apigenin, diosmetin, silibinin α, silibinin β, silydianin, silicristin, isosilibinin α, and isosilibinin β was prepared in methanol to the final concentration. Working solutions (5.0–20.0 μg/mL) were freshly prepared every day by the dilution of the standard stock solutions with methanol. All the solutions of each standard and mix standard, and 10 μL of the methanolic standardized extracts of *Cirsium* species and CH were injected in triplicate. All the chromatographic operations were performed at 25 °C.

### 4.5. Validation of the HPLC–DAD Method

The HPLC method was validated for its analytical specificity, linearity, precision, limit of detection (LOD), and limit of quantification (LOQ). The analytical specificity was confirmed to compare the retention times and UV spectra of the methanolic standardized extracts of *Cirsium* species and all the standards. The linearity of the detector response was validated by calculating the correlation coefficient (r^2^) for a linear regression equation from the calibration curve data for the above standards (5.0–20.0 μg/mL). To validate the precision parameter of the simultaneous analysis method, the intraday and interday variability were evaluated. The precision was expressed as the intraday and interday RSDs. The intraday and interday RSDs of the standards were assayed (three replicates) on the same day and on three sequential days, respectively. The limit of detection (LOD) was determined as the amount of standards corresponding to a signal-to-noise ratio of 3:1. The limit of quantitation (LOQ) was determined as the lowest concentration of the standards that could be quantified with acceptable precision and accuracy under the experimental conditions with a signal-to-noise ratio of at least 10.

### 4.6. DPPH-Radical-Scavenging Capacity

The DPPH-radical-scavenging capacity of the methanolic standardized extracts of *Cirsium* species and CH was determined by using a microtiter spectrophotometric reader (PowerWave X340, Bio-Tek Inc., Winooski, VT, USA), according to the method described in our previous reports [36]. One hundred and seventy-five microliters of 300 μM DPPH methanol solution was mixed with 25 μL of the methanolic standardized extracts, all the phytochemical ingredients, or the catechin standard. After 30 min of incubation at room temperature (RT), the absorbance of the reaction mixture at 517 nm was read. The scavenging capacity for the DPPH radical is expressed as milligrams of (+)-catechin equivalents per gram of the extracts (mg catechin/g sample), which is also called the catechin equivalents of DPPH scavenging capacity (CEDSC).

### 4.7. ABTS-Radical-Scavenging Activity

The ABTS-radical-scavenging activities of the methanolic standardized extracts of *Cirsium* species and CH were also determined by using a microtiter spectrophotometric PowerWave X340 reader, according to the method described in our previous reports [14]. The result of ABTS-radical-scavenging assay are represented by the amounts relative to the positive control Trolox, which is also called the Trolox equivalent antioxidant capacity (TEAC). Briefly, the ABTS radical was prepared from 8 mM ABTS solution and 8.4 mM potassium persulfate solution at a ratio of 2:1. Before use, the radical solution was freshly diluted with ethanol to reach an initial absorbance value (0.70 ± 0.05) at 734 nm. One hundred and seventy-five microliters of the diluted ABTS solution was mixed with 25 μL of the methanolic standardized extracts, all the phytochemical ingredients, or the Trolox standard. The TEAC values are expressed as mmol of Trolox equivalents per gram of the extracts (mmol Trolox/g sample).

### 4.8. Hypochlorite-Ion-Scavenging Assay

The hypochlorite-ion-scavenging activities of the methanolic standardized extracts of *Cirsium* species and CH were determined by using a microtiter spectrophotometric reader (PowerWave X340, Bio-Tek Inc., Winooski, VT, USA), modified from the method described by Hacke et al. [55]. Briefly, 150 μL of 0.004% sodium hypochlorite solution (in 50 mM sodium phosphate buffer solution (pH 7.4)) was mixed with 75 μL of the methanolic standardized extracts, all the phytochemical ingredients, or the Trolox standard. After 5 min of incubation at room temperature (RT), 25 μL of TMB was added into the reaction mixture. The absorbance at 652 nm was measured. The hypochlorite-ion-scavenging activities are expressed as mmol of Trolox equivalents per gram of the extracts (mmol Trolox/g sample).

### 4.9. Reducing-Power Assay

The FRAP assay was performed according to the method described in our previous report [36]. Briefly, 25 μL of the methanolic standardized extracts, all the phytochemical ingredients, or the Trolox standard was mixed with 25 μL of freshly prepared FRAP reagent (10 mM TPTZ in 40 mM HCl, 20 mM FeCl_3_, and 50 mM acetate buffer (pH 3.6)). The absorption of the reaction mixture at 595 nm was measured, and then, the FRAP values could be represented by the amount relative to the positive control Trolox, which is expressed as mmol of Trolox equivalents per gram of the extracts (mmol Trolox/g sample).

### 4.10. Statistical Analysis

All the results are expressed as the mean ± standard deviation (SD). The significance of differences was calculated using SPSS by one-way ANOVA followed by Scheffe’s test; values <0.05 were considered to be significant.

## 5. Conclusions

CJF showed the best radical-scavenging activities against all of the radicals and had the highest contents of total phenolics and phenylpropanoids among all the *Cirsium* materials. Among the collected parts of *Cirsium* species, better radical-scavenging activities and higher phenolic contents were found in the flower and aerial parts. There was a positive correlation between the total phenolic and phenylpropanoid contents and radical-scavenging activities. The radical-scavenging activities of the parts are closely related to their reducing power. These *Cirsium* species had different HPLC chromatograms. CJF contained silicristin, silydianin, silibinin, apigenin, and diosmetin, being especially high in silydianin. Silicristin had the best radical-scavenging activities against DPPH and ABTS; however, diosmetin had best reducing power and radical-scavenging activity against the hypochlorite ion. Flavonolignans (especially silicristin and silydianin) and diosmetin act synergistically in scavenging radicals; hence, they are major antioxidant ingredients of CJF.

## Figures and Tables

**Figure 1 molecules-26-03935-f001:**
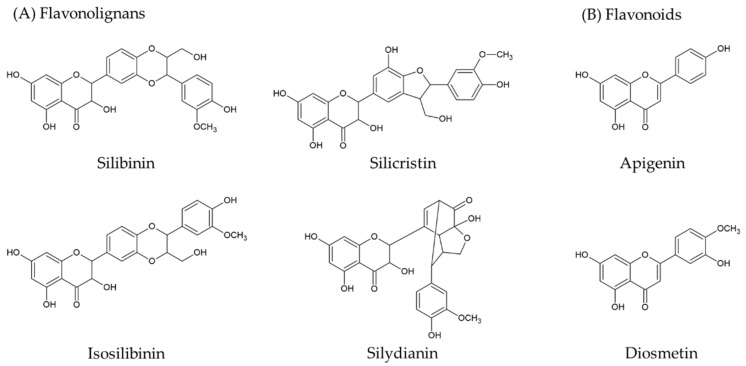
Structures of major ingredients ((**A**) flavonolignans and (**B**) flavonoids) of *Cirsium* species and Cirsii Herb.

**Figure 2 molecules-26-03935-f002:**
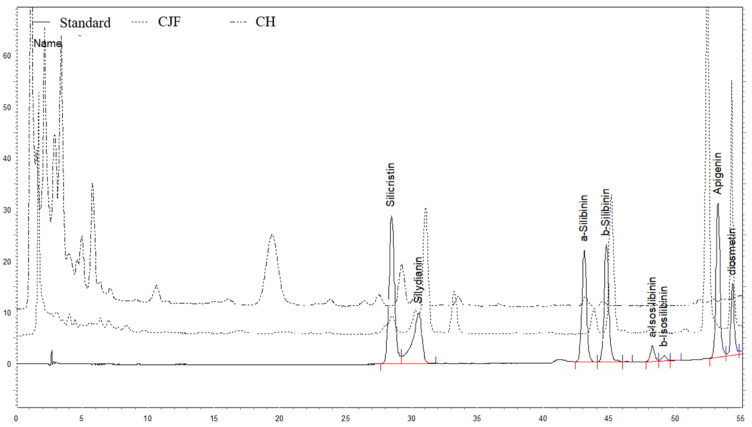
High-performance liquid chromatography (HPLC) chromatogram of standards, the methanolic standardized extracts of CJF, and Cirsii Herb at 280 nm. CJF, the flower part of *C. japonicum* DC. var. *australe*; CH, Cirsii Herb.

**Figure 3 molecules-26-03935-f003:**
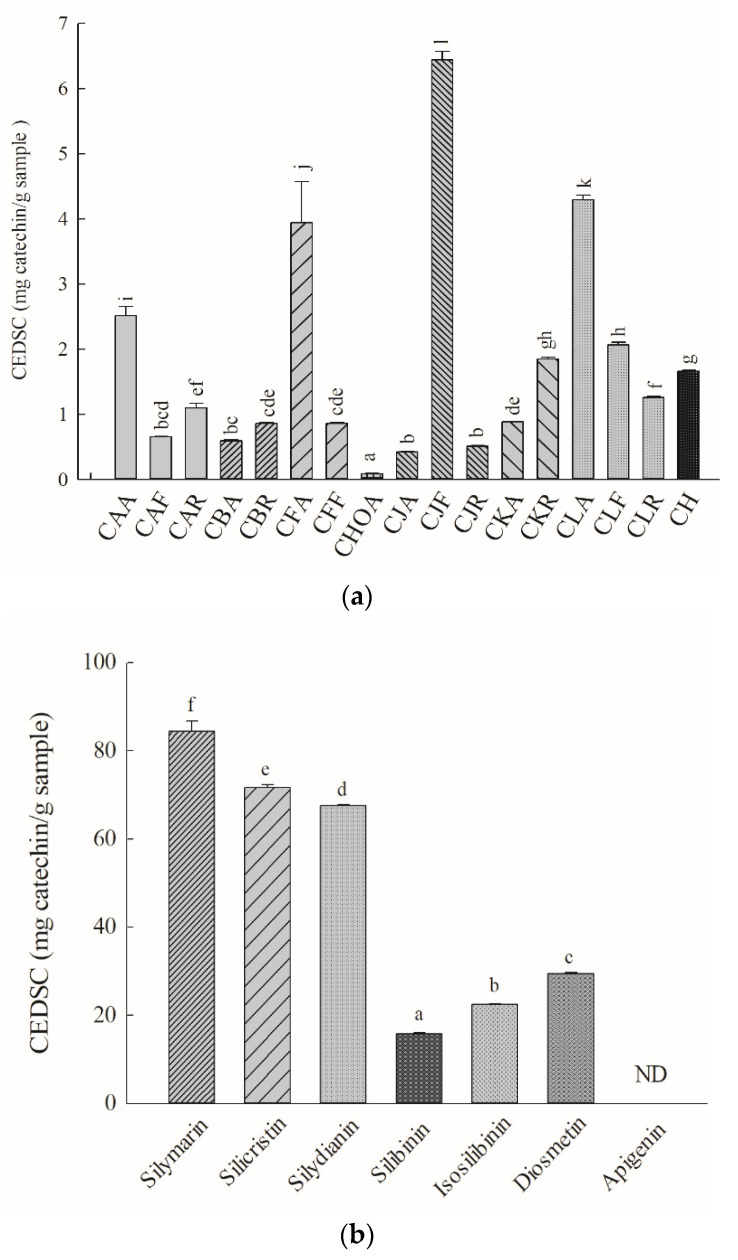
DPPH scavenging activities of the methanolic standardized extracts of *Cirsium* species and Cirsii Herb. (**a**) The methanolic standardized extracts of *Cirsium* species and CH; (**b**) all ingredients of the methanolic standardized extracts of *Cirsium* species and CH. Data are expressed as means ± SD (n = 3). ND: non-detected. CAA, the aerial part of *C. arisanense*; CAF, the flower part of *C. arisanense*; CAR, the radix part of *C. arisanense*; CBA, the aerial part of *C. brevicaule*; CBR, the radix part of *C. brevicaule*; CEDSC. catechin equivalent of DPPH scavenging capacity; CFA, the aerial part of *C. ferum*; CFF, the flower part of *C. ferum*; CHOA, the aerial part of *C. hosokawae*; CJA, the aerial part of *Cirsium japonicum* DC. var. *australe*; CJF, the flower part of *C. japonicum* DC. var. *australe*; CJR, the radix part of *C. japonicum* DC. var. *australe*; CKA, the aerial part of *C. kawakamii*; CKR, the radix part of *C. kawakamii*; CLA, the aerial part of *C. lineare*; CLF, the flower part of *C. lineare*; CLR, the radix part of *C. lineare*; CH, Cirsii Herb; DPPH, 1,1-diphenyl-2-picrylhydrazyl. Bars with different letters indicate statistical significances, *p* < 0.05.

**Figure 4 molecules-26-03935-f004:**
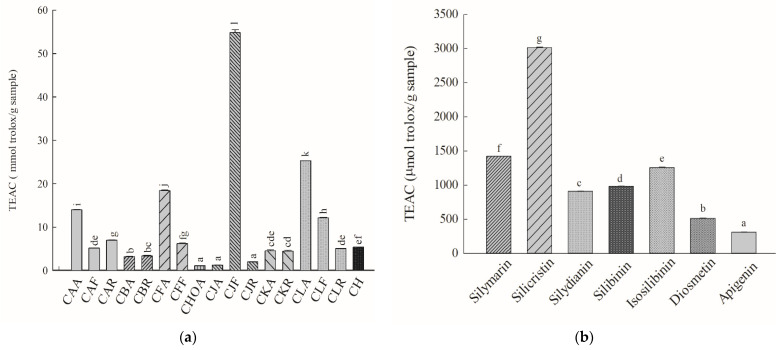
ABTS scavenging activities of the methanolic standardized extracts of *Cirsium* species and Cirsii Herb. (**a**) The methanolic standardized extracts of *Cirsium* species and CH; (**b**) all ingredients of the methanolic standardized extracts of *Cirsium* species and CH. Data are expressed as means ± SD (n = 3). ABTS, 2,2′-azino-bis-3-ethylbenzthiazoline-6-sulfonic acid; CAA, the aerial part of *C. arisanense*; CAF, the flower part of *C. arisanense*; CAR, the radix part of *C. arisanense*; CBA, the aerial part of *C. brevicaule*; CBR, the radix part of *C. brevicaule*; CFA, the aerial part of *C. ferum*; CFF, the flower part of *C. ferum*; CHOA, the aerial part of *C. hosokawae*; CJA, the aerial part of *C. japonicum* DC. var. *australe*; CJF, the flower part of *C. japonicum* DC. var. *australe*; CJR, the radix part of *C. japonicum* DC. var. *australe*; CKA, the aerial part of *C. kawakamii*; CKR, the radix part of *C. kawakamii*; CLA, the aerial part of *C. lineare*; CLF, the flower part of *C. lineare*; CLR, the radix part of *C. lineare*; CH, Cirsii Herb. Bars with different letters indicate statistical significances, *p* < 0.05.

**Figure 5 molecules-26-03935-f005:**
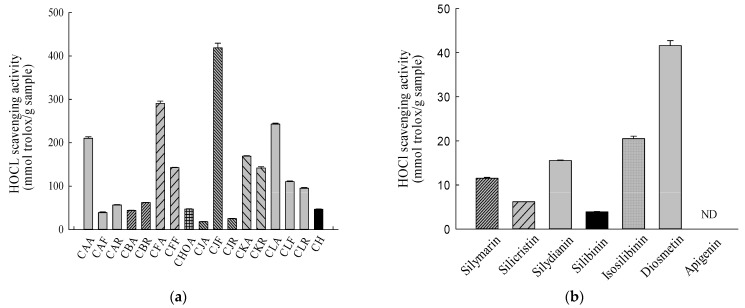
HClO-scavenging activities of the methanolic standardized extracts of *Cirsium* species and Cirsii Herb. (**a**) The methanolic standardized extracts of *Cirsium* species and CH; (**b**) all ingredients of the methanolic standardized extracts of *Cirsium* species and CH. Data are expressed as means ± SD (n = 3). ND: non-detected. CAA, the aerial part of *C. arisanense*; CAF, the flower part of *C. arisanense*; CAR, the radix part of *C. arisanense*; CBA, the aerial part of *C. brevicaule*; CBR, the radix part of *C. brevicaule*; CFA, the aerial part of *C. ferum*; CFF, the flower part of *C. ferum*; CHOA, the aerial part of *C. hosokawae*; CJA, the aerial part of *C. japonicum* DC. var. *australe*; CJF, the flower part of *C. japonicum* DC. var. *australe*; CJR, the radix part of *C. japonicum* DC. var. *australe*; CKA, the aerial part of *C. kawakamii*; CKR, the radix part of *C. kawakamii*; CLA, the aerial part of *C. lineare*; CLF, the flower part of *C. lineare*; CLR, the radix part of *C. lineare*; CH, Cirsii Herb; HOCl, hypochlorite acid. Bars with different letters indicate statistical significances, *p* < 0.05.

**Figure 6 molecules-26-03935-f006:**
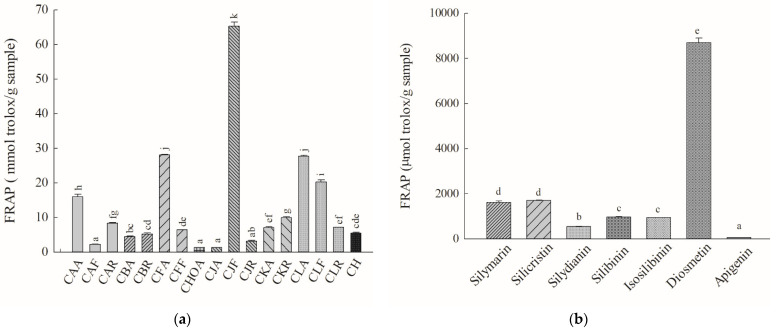
Reducing power of the methanolic standardized extracts of *Cirsium* species and Cirsii Herb. (**a**) The methanolic standardized extracts of *Cirsium* species and CH; (**b**) all ingredients of the methanolic standardized extracts of *Cirsium* species and CH. Data are expressed as means ± SD (n = 3). CAA, the aerial part of *C. arisanense*; CAF, the flower part of *C. arisanense*; CAR, the radix part of *C. arisanense*; CBA, the aerial part of *C. brevicaule*; CBR, the radix part of *C. brevicaule*; CFA, the aerial part of *C. ferum*; CFF, the flower part of *C. ferum*; CHOA, the aerial part of *C. hosokawae*; CJA, the aerial part of *C. japonicum* DC. var. *australe*; CJF, the flower part of *C. japonicum* DC. var. *australe*; CJR, the radix part of *C. japonicum* DC. var. *australe*; CKA, the aerial part of *C. kawakamii*; CKR, the radix part of *C. kawakamii*; CLA, the aerial part of *C. lineare*; CLF, the flower part of *C. lineare*; CLR, the radix part of *C. lineare*; CH, Cirsii Herb; FRAP, ferric reducing antioxidant power. Bars with different letters indicate statistical significances, *p* < 0.05.

**Table 1 molecules-26-03935-t001:** The contents of total phenolics, total flavonoids, and total phenylpropanoids of *Cirsium* species and Cirsii Herb.

Species	Samples	Total Phenolics (mg Catechin/g Sample)	Total Flavonoids (mg Quercetin/g Sample)	Total Phenylpropanoids (mg Verbascoside/g Sample)
CA	CAA	2.79 ± 0.07 ^g^	4.07 ± 0.19 ^bcde^	6.39 ± 0.10 ^e^
	CAF	0.68 ± 0.01 ^d^	0.11 ± 0.04 ^a^	1.02 ± 0.73 ^ab^
	CAR	1.33 ± 0.01 ^e^	2.42 ± 0.10 ^abc^	3.46 ± 0.13 ^d^
CB	CBA	0.75 ± 0.02 ^d^	2.48 ± 0.89 ^abc^	1.08 ± 0.06 ^ab^
	CBR	1.34 ± 0.01 ^e^	0.47 ± 0.01 ^a^	1.81 ± 0.01 ^abc^
CF	CFA	3.10 ± 0.13 ^g^	4.98 ± 0.27 ^de^	12.29 ± 0.48 ^f^
	CFF	0.35 ± 0.02 ^ab^	2.49 ± 0.10 ^abc^	3.32 ± 0.18 ^d^
CHO	CHOA	0.46 ± 0.00 ^bc^	0.27 ± 0.03 ^a^	3.10 ± 0.08 ^cd^
CJ	CJA	0.35 ± 0.01 ^ab^	0.99 ± 0.59 ^ab^	0.31 ± 0.00 ^a^
	CJF	5.23 ± 0.20 ^i^	2.24 ± 0.36 ^abc^	29.73 ± 0.72 ^g^
	CJR	0.08 ± 0.00 ^a^	1.77 ± 0.29 ^abc^	1.67 ± 0.08 ^abc^
CK	CKA	1.70 ± 0.04 ^f^	2.62 ± 2.82 ^abc^	5.49 ± 0.22 ^e^
	CKR	1.69 ± 0.02 ^f^	0.60 ± 0.21 ^a^	2.88 ± 0.05 ^cd^
CL	CLA	3.95 ± 0.12 ^h^	6.37 ± 0.19 ^e^	11.87 ± 0.49 ^f^
	CLF	2.91 ± 0.03 ^g^	4.66 ± 0.12 ^cde^	6.53 ± 0.17 ^e^
	CLR	1.08 ± 0.01 ^e^	1.36 ± 0.04 ^abc^	2.25 ± 0.09 ^bcd^
CH	CH	0.63 ± 0.08 ^bc^	2.10 ± 0.34 ^abc^	2.23 ± 0.10 ^bcd^

Data are expressed as means ± SD (n = 3). CAA, the aerial part of *C. arisanense*; CAF, the flower part of *C. arisanense*; CAR, the radix part of *C. arisanense*; CBA, the aerial part of *C. brevicaule*; CBR, the radix part of *C. brevicaule*; CFA, the aerial part of *C. ferum*; CFF, the flower part of *C. ferum*; CHOA, the aerial part of *C. hosokawae*; CJA, the aerial part of *C. japonicum* DC. var. *australe*; CJF, the flower part of *C. japonicum* DC. var. *australe*; CJR, the radix part of *C. japonicum* DC. var. *australe*; CKA, the aerial part of *C. kawakamii*; CKR, the radix part of *C. kawakamii*; CLA, the aerial part of *C. lineare*; CLF, the flower part of *C. lineare*; CLR, the radix part of *C. lineare*; CH, Cirsii Herb. Different letters represent significant differences, *p* < 0.05.

**Table 2 molecules-26-03935-t002:** Linearity and Sensitivity Data of All Standards.

Standard	Linear Equation	Coefficient of Determination (r^2^)	Linearity Range (μg/mL)	LOD (μg/mL)	LOQ (μg/mL)
Silicristin	y = 23,969x − 2132	0.9987	4.91–19.96	0.31	0.93
Silydianin	y = 19,399x − 35,932	0.9987	5.08–20.04	0.64	1.91
Silibinin α	y = 10,480x − 12,673	0.9995	5.05–20.02	1.21	3.63
Silibinin β	y = 11,152x − 12,022	0.9989	5.09–20.04	1.21	3.63
Isosilibinin α	y = 9672x − 16,561	0.9985	4.92–19.96	1.33	4.00
Isosilibinin β	y = 2799x − 7293	0.9989	5.00–20.00	1.33	4.00
Apigenin	y = 19,209x − 7119	0.9980	4.90–19.98	0.67	2.40
Diosmetin	y = 16,308x − 18,966	0.9994	5.05–20.02	0.80	2.00

**Table 3 molecules-26-03935-t003:** Intraday and Interday Variabilities of All Standards.

Standard	Concentration (μg/mL)	Intraday (n = 3)		Interday (n = 3)	
		Mean ± SD	RSDs (%)	Mean ± SD	RSDs (%)
Silicristin	5.0	4.91 ± 0.02	0.27	4.92 ± 0.24	2.98
	10.0	10.13 ± 0.03		10.26 ± 0.14	
	20.0	19.96 ± 0.01		19.86 ± 0.20	
Silydianin	5.0	5.08 ± 0.18	2.23	5.26 ± 0.19	1.66
	10.0	9.89 ± 0.27		9.83 ± 0.06	
	20.0	20.04 ± 0.09		20.19 ± 0.10	
Silibinin α	5.0	5.05 ± 0.04	0.46	5.07 ± 0.10	2.66
	10.0	9.93 ± 0.06		10.43 ± 1.00	
	20.0	20.02 ± 0.02		20.17 ± 0.02	
Silibinin β	5.0	5.09 ± 0.13	1.57	4.95 ± 0.21	1.78
	10.0	9.87 ± 0.19		10.00 ± 0.32	
	20.0	20.04 ± 0.06		20.19 ± 0.14	
Isosilibinin α	5.0	4.92 ± 0.05	0.66	5.20 ± 0.14	1.76
	10.0	10.11 ± 0.08		10.12 ± 0.22	
	20.0	19.96 ± 0.03		20.21 ± 0.22	
Isosilibinin β	5.0	5.00 ± 0.04	0.50	4.87 ± 0.18	1.86
	10.0	10.00 ± 0.06		9.85 ± 0.25	
	20.0	20.00 ± 0.02		20.13 ± 0.05	
Apigenin	5.0	4.92 ± 0.05	2.03	4.88 ± 0.26	2.74
	10.0	10.24 ± 0.18		10.28 ± 0.16	
	20.0	19.92 ± 0.06		19.77 ± 0.30	
Diosmetin	5.0	5.05 ± 0.05	0.66	5.04 ± 0.03	2.20
	10.0	9.93 ± 0.08		9.96 ± 0.26	
	20.0	20.02 ± 0.03		19.88 ± 0.12	

RSD, relative standard deviation; SD, standard deviation.

**Table 4 molecules-26-03935-t004:** The contents of major ingredients in *Cirsium* species in Taiwan.

Samples	Silicristin (mg/g Sample)	Silydianin (mg/g Sample)	Silibinin α (mg/g Sample)	Silibinin β (mg/g Sample)	Isosilibinin β (mg/g Sample)	Apigenin (mg/g Sample)	Diosmetin (mg/g Sample)
CAA	ND	ND	0.76 ± 0.01	ND	ND	ND	ND
CBA	ND	ND	ND	0.34 ± 0.05	ND	ND	ND
CFA	ND	ND	3.15 ± 0.02	4.62 ± 0.04	ND	ND	ND
CFF	ND	ND	1.19 ± 0.04	4.13 ± 0.12	ND	ND	ND
CJF	0.66 ± 0.03	9.13 ± 0.30	0.48 ± 0.04	1.11 ± 0.14	ND	5.56 ± 0.09	2.82 ± 0.10
CKA	ND	ND	0.03 ± 0.00	0.24 ± 0.00	ND	0.08 ± 0.00	ND
CKR	ND	ND	ND	0.24 ± 0.00	ND	ND	ND
CLA	ND	ND	ND	ND	ND	ND	1.39 ± 0.01
CLF	ND	ND	ND	ND	0.33 ± 0.01	0.62 ± 0.02	1.71 ± 0.05
CH	ND	ND	0.01 ± 0.00	0.01 ± 0.00	ND	ND	ND

Data are expressed as means ± SD (n = 3). ND: non-detected. CAA, the aerial part of *C. arisanense*; CBA, the aerial part of *C. brevicaule*; CFA, the aerial part of *C. ferum*; CFF, the flower part of *C. ferum*; CJF, the flower part of *C. japonicum* DC. var. *australe*; CKA, the aerial part of *C. kawakamii*; CKR, the radix part of *C. kawakamii*; CLA, the aerial part of *C. lineare*; CLF, the flower part of *C. lineare*; CH, Cirsii Herb.

**Table 5 molecules-26-03935-t005:** Pearson correlation coefficients (r) between parameters describing the contents of total phenolics and phenylpropanoids and different antioxidant activities of the methanolic standardized extracts of *Cirsium* species and Cirsii Herb.

	DPPH	ABTS	ClO^−^	FRAP
ABTS	0.949			
ClO^−^	0.925	0.891		
FRAP	0.961	0.986	0.913	
Total Phenolics	0.940	0.897	0.900	0.923
Total phenylpropanoids	0.938	0.979	0.906	0.980

ABTS, 2,2′-azino-bis-3-ethylbenzthiazoline-6-sulfonic acid; DPPH, 1,1-diphenyl-2-picrylhydrazyl; FRAP, ferric reducing antioxidant power; ClO^−^, hypochlorite ion.

**Table 6 molecules-26-03935-t006:** Abbreviations and Collection Places of Taiwanese *Cirsium* species and Cirsii Herb.

Species	Part	Abbreviation	Collection Place
*C. arisanense* Kitam. (CA)	aerial	CAA	Yuanfeng, Nantou (Altitude 2500–2800 m)
	flower	CAF	
	radix	CAR	
*C. brevicaule* A. Gray (CB)	aerial	CBA	Tianwei, Changhua (Altitude 25–30 m)
	radix	CBR	
*C. ferum* Kitam. (CF)	aerial	CFA	Yuanfeng, Nantou (Altitude 2500–2800 m)
	flower	CFF	
*C. hosokawae* Kitam. (CHO)	aerial	CHOA	Alishan, Chiayi (Altitude 1300–1500 m)
*C. japonicum* DC. *var. australe* Kitam. (CJ)	aerial	CJA	Alishan, Chiayi (Altitude 1300–1500 m)
	flower	CJF	
	radix	CJR	
*C. kawakamii* Hayata (CK)	aerial	CKA	Alishan, Chiayi (Altitude 1300–1500 m)
	radix	CKR	
*C. lineare* (Thunb.) Sch. Bip. (CL)	aerial	CLA	Yuanfeng, Nantou (Altitude 2500–2800 m)
	flower	CLF	
	radix	CLR	
Cirsii Herb (CH)	Herb	CH	Taichung (Altitude 25–30 m)

**Table 7 molecules-26-03935-t007:** HPLC gradient program conditions of methanolic standardized extracts of *Cirsium* species and CH.

Time (min)	Solvent A (%)	Solvent B (%)
0–15	90	10
15–25	70	30
25–35	55	45
35–45	35	65
45–50	0	100
50–55	0	100
55–65	90	10

Solvent A: phosphoric acid: methanol: water = 0.5:35:65. Solvent B: phosphoric acid: methanol: water = 0.5:70:30.

## Data Availability

Not applicable.

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
