# Peer review of "Antioxidant Effects and Phytochemical Properties of Seven Taiwanese Cirsium Species Extracts"

_molecules, 2021, doi:10.3390/molecules26133935_

Round 1
Reviewer 1 Report
Molecules-1240682
Considering the title of manuscript, the reviewed material concerns the method developed for the evaluation of antioxidant properties and phytochemical characterisation of extracts obtained from seven Cirsium species. Mentioned species have been recorded in the Flora of Taiwan. At most, various morphological parts of plants have been taken into account.
Radical scavenging activities against 1,1‐diphenyl‐2‐picrylhydrazyl (DPPH), 2,2′‐azino‐bis(3‐ethylbenzothiazoline‐6‐sulphonic acid) (ABTS), and hypochlorite ion and their phenolic contents with spectrophotometric method have been compared. For the phytochemical characterisation of extracts some profiles obtained by use of high‐performance liquid chromatography–photodiode array detector (HPLC‐DAD) were presented.
The mentioned idea of investigations is not particularly original and several major improvements are still needed in order to achieve a relevant manuscript. After reading this manuscript, some comments and qualifications came to mind as follows.
- Beginning with the abstract, there is no significant data on the results obtained. A big mistake is the lack of quantitative analysis results. The content should change significantly.
- Likewise, the introduction part requires a complete revision. There is no information about the research methodology usually applied for the characterisation of Cirsium species. Moreover, in this type of works, personal pronouns are not used.
- This study does not assess the main factors and interactions among the variables, in sample preparation method and chromatographic analysis. The Authors should solve or justify properly this lack.
- Moreover, results concerns HPLC analysis are not at all sufficient for the original research. Lack of the proper qualitative analysis, in this reason the paper must be supplied with a particular statistical analysis (validation). All calibration data including linearity with correlation coefficient of the calibration curves, limits of detection (LOD), precision as a relative standard deviation (RSD) should be presented. Some literature data have been cited by Authors, which is not well appreciated.
- Any scientific work should contain important elements of novelty and a new approach to the discussed problems. This aspect is certainly missing from the reviewed work. What are the significant differences from other scientific works in this field? Is it possible to identify new compounds in the studied extracts, not described in the literaturę, so far? Please provide a method for the identification of unknown compounds, as well as provide a complete database for the phytochemical characterisation of mentioned Cirsium species.
Summary:
The subject of the manuscript falls within the scope of Molecules. Nevertheless, due to numerous doubts as well as the lack of essential elements of scientific work, the reviewed manuscript is recommended to publish after major revision.
Author Response
- Beginning with the abstract, there is no significant data on the results obtained. A big mistake is the lack of quantitative analysis results. The content should change significantly.
Answer: Thanks for the suggestion. We have rewritten the “Abstract” section according to this comment.
- The introduction part requires a complete revision
Answer: Thanks for the suggestion. We have rewritten the “Introduction” section according to this comment.
- This study does not assess the main factors and interactions among the variables, in sample preparation method and chromatographic analysis.
Answer: Thanks for the suggestion. The main purposes of this work are to confirm that different Cirsium species contain different ingredients and have different degrees of radical scavenging activities, and to distinguish different Cirsium species by HPLC method and understand the synergic effects of the major ingredients.
- Lack of the proper qualitative analysis, in this reason the paper must be supplied with a particular statistical analysis (validation). All calibration data including linearity with correlation coefficient of the calibration curves, limits of detection (LOD), precision as a relative standard deviation (RSD) should be presented.
Answer: Thanks for the suggestion. We have provided the validation of HPLC method including linearity with correlation coefficient of the calibration curves, limits of detection (LOD), precision as a relative standard deviation (RSD). These results are presented in the “2.2. Validation of HPLC-DAD method and The Phytochemical Profiles of Cirsium species” of “Results” section. The data are shown in the Table 2. This method is descripted in the “4.5. Validation of HPLC-DAD method” of “Materials and Methods” section.
- Any scientific work should contain important elements of novelty and a new approach to the discussed problems. This aspect is certainly missing from the reviewed work. What are the significant differences from other scientific works in this field?
Answer: Thanks for the suggestion. We have rewritten the “Introduction” section and described the purposes and significance of this work.

Reviewer 2 Report
Dear Authors,
After the review process, I have several comments: you should include numerical data in the abstract; you should check the paper and made the necessary correction in the text, e.g. Cirsium should be italic; you should include statistical data in the figures; you should comment their results based on the bioavailability process, that is one of the most important aspects for antioxidant effects. For example, you should comment on the extract efficiency based on the bioactive potential of functional products and the bioavailability of phenolic compounds. In addition, the correlation with new lab methods will improve the bioavailability and bioactivities of polyphenols extracts after in vitro digestion, an important aspect after product administration.
Best regards.
Author Response
- you should include numerical data in the abstract
Answer: Thanks for the suggestion. We have rewritten the “Abstract” section according to this comment.
- Cirsium should be italic
Answer: Thanks for the suggestion. We have corrected them.
- you should include statistical data in the figures
Answer: Thanks for the suggestion. We have added the statistical data in the figures.
- you should comment their results based on the bioavailability process, that is one of the most important aspects for antioxidant effects. For example, you should comment on the extract efficiency based on the bioactive potential of functional products and the bioavailability of phenolic compounds.
Answer: Thanks for the suggestion. We have added the related description and literatures in the “Discussion” section.

Round 2
Reviewer 2 Report
ZDear Authors,
No other comments compared to the first review. You included general comments on bioavailability, but which is the effect of microbiota bioactivity?
This aspect is essential in product efficiency?
Best regards.
Author Response
- which is the effect of microbiota bioactivity? This aspect is essential in product efficiency?
Answer: Thanks for the suggestion. We have added the related description (line 489-518, page 14-15) and literatures (Ref. 47-54) in the “Discussion” section.
